# KVLINK: Accelerating Large Language Models via Efficient KV Cache Reuse

**Jingbo Yang**[*]
Department of Computer Science
UC Santa Barbara
jingbo@ucsb.edu

**Bairu Hou**[*]
Department of Computer Science
UC Santa Barbara
bairu@ucsb.edu

**Wei Wei**
Center for Advanced AI
Accenture
wei.h.wei@accenture.com

**Yujia Bao**
Center for Advanced AI
Accenture
yujia.bao@accenture.com

**Shiyu Chang**
Department of Computer Science
UC Santa Barbara
chang87@ucsb.edu

## Abstract

We describe KVLINK, an approach for efficient key-value (KV) cache reuse in large language models (LLMs). In many LLM applications, different inputs can share overlapping context, such as the same retrieved document appearing in multiple queries. However, the LLMs still need to encode the entire context for each query, leading to redundant computation. In this paper, we investigate a new strategy to eliminate such inefficiency, where the KV cache of each document is precomputed independently. During inference, the KV caches of retrieved documents are concatenated, allowing the model to reuse cached representations instead of recomputing them. To mitigate the performance degradation when using KV caches computed independently for each document, KVLINK introduces two key techniques: adjusting positional embeddings of the KV cache at inference to match the global position after concatenation, and using trainable special tokens to restore self-attention across independently encoded documents. Experiments across 7 datasets demonstrate that KVLINK improves question answering accuracy by an average of 4% over state-of-the-art methods. Furthermore, by leveraging precomputed KV caches, our approach reduces time-to-first-token by up to 96% compared to standard LLM inference, making it a scalable and efficient solution for context reuse. Additionally, KVLINK can be combined with KV cache compression to further save cache loading and storage overhead while outperforming the baselines. Code is available at `https://github.com/UCSB-NLP-Chang/KVLink`.

## 1  Introduction

Large language models have demonstrated impressive capabilities across a broad array of applications—many of which involve processing contexts naturally divided into multiple segments. For example, in retrieval-augmented generation (RAG) [1–3], each retrieved document forms a distinct

---

[*]Equal contribution.

39th Conference on Neural Information Processing Systems (NeurIPS 2025).

context chunk, while in multi-agent conversation scenarios [4, 5], outputs from different agents serve as separate segments. However, conventional architectures require LLMs to encode the entire concatenated context as a single unit before generating a response. This approach incurs high prefilling costs for long contexts and prevents the model from separately encoding and reusing precomputed representations (*i.e.*, key-value states) for each segment. Consider RAG: for every query, the LLM encodes a large collection of retrieved documents. When different queries share common documents, the model redundantly re-encodes these identical texts, even though their content remains unchanged.

This inefficiency motivates us to explore an alternative strategy. Instead of re-encoding the entire concatenated context for every query, we propose precomputing the key-value (KV) states for each document or text segment independently, then reusing these precomputed states during inference. By encoding each segment (*e.g.*, each retrieved document) separately and concatenating their KV states as needed, we can eliminate redundant computations and significantly improve efficiency. However, naively encoding each document independently and concatenating their KV caches during inference can lead to performance degradation due to train-test discrepancies. Prior work [6–8] has reported up to a 35% relative decrease in accuracy on QA tasks, which results from the discrepancy of position embeddings and missing cross attention between retrieved documents. To overcome this challenge, we introduce KVLINK, an approach designed to bridge the gap between separately encoded segments and restore self-attention across documents. KVLINK introduces two key enhancements: ❶ KV cache positional re-encoding. We adjust positional embeddings during inference to ensure that the stored KV caches align correctly with the positions required for a given query. ❷ Trainable cross-segment special tokens. To effectively "link" independently encoded segments, KVLINK appends a small set of trainable tokens between each segment's precomputed KV states before concatenation. The KV representations for these tokens are computed during inference. While the documents are independently encoded into KV cache, the link tokens can attend to all the preceding tokens, which helps restore self-attention across segments while introducing only minimal computational overhead.

We validate the effectiveness and efficiency of our method through comprehensive experiments across diverse question-answering and text summarization datasets. We evaluate three backbone LLMs with different model scales, including `Llama-3.2-1B`, `Llama-3.2-3B`, and `Llama-3.1-8B`, showing that our approach consistently outperforms state-of-the-art baselines [6, 7, 9]. For example, KVLINK surpasses the best baseline by 6.6% on Natural Question [10] and 7.3% on HotpotQA [11]. Even compared to the upper-bound model (*i.e.*, LLM using conventional full encoding for inference), our method sacrifices only minimal performance while reducing the time-to-first-token latency by up to 96% by efficiently reusing precomputed KV caches.

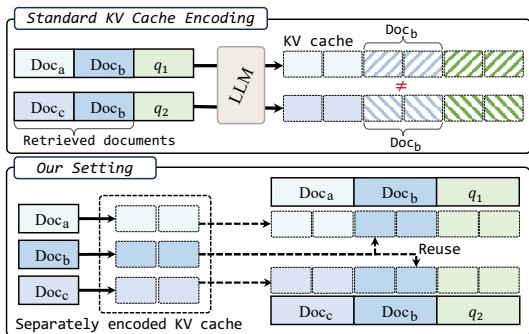

Figure 1: Standard approach (top) encodes the KV cache of each document conditioned on preceding tokens, resulting in redundant and nonreusable KV cache encoding for shared documents (*e.g.*, $\text{Doc}_b$). In contrast, our setting (bottom) encodes documents separately, allowing KV cache reuse across queries.

Additionally, we investigate the integration of KVLINK with KV cache compression techniques, as storing precomputed KV caches can incur substantial storage overhead. By incorporating methods such as LLMLINGUA [12] and ANLLMS [13], KVLINK consistently outperforms baseline approaches. These results demonstrate its strong potential for practical, large-scale deployment.

## 2 Methodology

### 2.1 Problem Formulation

We aim to address the problem of reusing the key-value (KV) cache in LLMs without having to repeatedly encode the same context segments. In many applications, the same documents or context

segments appear across different inference queries, yet current LLM pipelines redundantly re-encode these segments every time.

Consider the problem of retrieval augmented generation as an example, where the input question is denoted as $q$ and we retrieve $N$ documents for each question. These documents are concatenated with the query, forming the complete LLM input.

In the standard LLM inference pipeline, the entire input is passed into the LLM as a single contiguous sequence. As shown in the top portion of Figure 1, the resulting KV cache for each document is entangled with its preceding documents. Consequently, even if $\text{Doc}_b$ has already been encoded into KV cache when processing $q_1$, the resulting KV cache cannot be directly reused when processing $q_2$ because it was conditioned on the preceding documents in $q_1$.

To address this, we consider a scenario where the KV cache for every document in the knowledge base is pre-computed in a *context-free* manner, as illustrated in the bottom of Figure 1. Specifically, for each document in the knowledge base, we feed only that document's tokens into the LLM and record the resulting KV cache. At inference time, after we retrieve a set of documents for a given query, we concatenate their pre-computed KV caches. This design allows us to reuse the same cache for overlapping documents across different queries, thereby eliminating redundant computations.

However, the above approach often yields noticeable performance degradation, as LLMs are typically trained on fully concatenated sequences and each token attends to the preceding context. When we instead encode each document in a context-free manner, the model loses cross-document dependencies. Previous work also empirically demonstrates up to a 35% relative decrease in accuracy in question-answering tasks when each retrieved document is encoded into KV cache separately [7]. Our method aims to enable the LLM to produce high-quality outputs when the KV cache of each document is pre-computed independently by introducing two key components, ❶ KV cache positional re-encoding and ❷ cross-document reconnection with link tokens.

## 2.2 KV Cache Positional Re-encoding

The first issue with separately encoding documents is the position mismatch during inference. Modern LLMs typically use Rotary Positional Encoding (RoPE) [14], where each token is assigned a distinct positional embedding based on its position in the full sequence. However, when documents are encoded independently, tokens are indexed within their own document, ignoring their actual position in the full concatenated input. Take Figure 1 as an example. Since we pre-compute the KV cache for each document separately, the second token in $\text{Doc}_b$ is assigned position index 2 when computing its KV cache. However, when we concatenate $\text{Doc}_b$ with $\text{Doc}_a$ during inference (e.g., when processing query $q_1$), the actual position of that token should be $|\text{Doc}_a| + 2$ where $|\text{Doc}_a|$ is the length of the first document. Since the KV cache of $\text{Doc}_b$ was precomputed without awareness of this global position shift, the LLM applies incorrect positional embeddings, leading to erroneous attention computations.

To address this, we decouple the key-value states from the positional embeddings when storing them. We still apply rotary position encoding to tokens when encoding each segment for local self-attention. However, we exclude those positional transformations before saving the segment KV caches.

More specifically, we denote the hidden state of a token at position $i$ at a particular transformer layer as $\boldsymbol{x}_i \in \mathbb{R}^d$, where $d$ is the hidden dimension. The key vector is computed as $R_i W_k \boldsymbol{x}_i$ and the value vector is computed as $W_v \boldsymbol{x}_i$. Here $W_{\{k,v\}} \in \mathbb{R}^{d \times d}$ represents the weight matrices that project the hidden states into the key and value spaces, and $R_i \in [-1, 1]^{d \times d}$ is the position-dependent rotation matrix used in RoPE. Because RoPE directly encodes positional information via these rotations, the KV cache can be stored without positional embeddings, *i.e.*, $W_{\{k,v\}} \boldsymbol{x}_i$. At inference time, the key-value states of all documents are concatenated, and we apply the global rotary embedding for the KV states of each token appropriate to its location in the full sequence. This operation only introduces negligible time, ensuring the efficiency of our method. This is also consistent with previous methods [6, 8], which adjust the position encoding of separately encoded KV cache during inference.

## 2.3 Cross-Document Reconnection with Link Tokens

When documents are encoded and cached independently, tokens in later documents can no longer attend to those in earlier ones. This creates a gap between our inference scheme and standard LLM training and can potentially limit the model performance. To mitigate this issue, we design

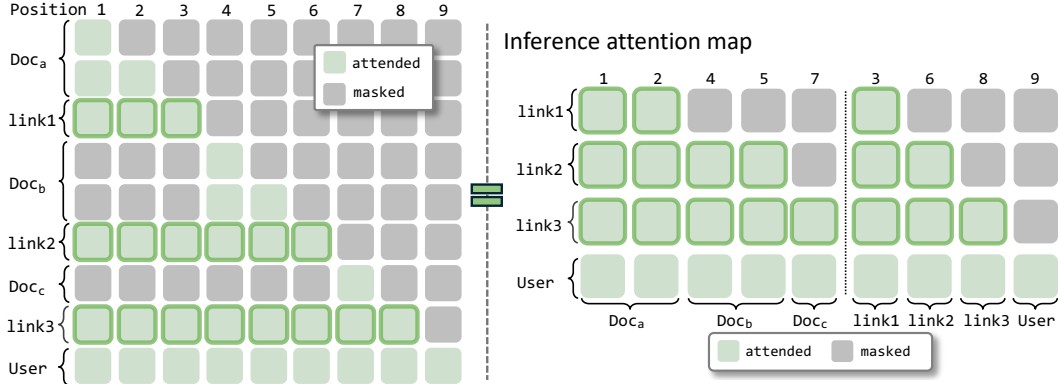

Figure 2: Left: the attention map for all tokens. The `link1` token attends only to the tokens in $\text{Doc}_a$; `link2` attends to the tokens in $\text{Doc}_a$ and $\text{Doc}_b$ and `link2`; and `link3` attends to the tokens in all reused contexts and other link tokens. Right: the attention map for three link tokens and the first user input token during inference. These two attention maps are identical.

a novel mechanism to restore the self-attention connection across segments while preserving the computational efficiency of KV cache reuse. Specifically, KVLINK introduces a set of trainable "link tokens". For every document $c$ with length $L$, we append $K$ (*e.g.*, $K = 5$) link tokens:

$$c = \left(c_1, \ldots, c_L, c_{\text{link1}}, \ldots, c_{\text{link5}}\right),$$

where $c_1, \ldots, c_L$ are the tokens within the original document. A customized attention map ensures that the link tokens of each document can attend to (i) all tokens (including the link tokens) in the preceding documents and (ii) tokens within the current document. These tokens serve as an interface between segments, allowing the model to reconstruct dependencies that would otherwise be lost due to independent encoding.

Figure 2 illustrates this mechanism, where one link token per document ($K = 1$) is append (token 3, 6, and 8). The tokens within a document can only maintain local causal attention, meaning they can only attend to earlier tokens within the same document. Each link token can attend to all preceding tokens. Therefore, different documents are implicitly connected through these link tokens during inference. For example, token 6 (the link token of document 2) attends to the first two documents and token 8 (the link token of document 3) attends to token 6, thus mixing the information across all retrieved documents. After the retrieved documents, the question and user tokens follow standard causal self-attention. During training, we fine-tune the LLM with this attention mechanism, jointly optimizing both the model and the newly introduced link tokens.

Figure 2 illustrates our inference strategy, which avoids recomputing KV caches for retrieved documents. In practice, we first load and concatenate the precomputed KV caches for all retrieved documents. Since these caches were precomputed, they do not need to be re-encoded. We then append only the newly introduced link tokens (tokens 3, 6, and 8, corresponding to the three documents) to the end of the reordered sequence. A forward pass is performed on these tokens using the customized attention map, ensuring that their attention behavior matches the training phase.

## 2.4 Compressed KV Cache Linking

Although reusing pre-computed KV caches can greatly accelerate inference, it may introduce considerable overhead in storing these caches in a database and loading them onto the GPU. This overhead presents a major challenge for integrating cache reuse into real-world RAG deployments. For example, a 1,000-token document stored as UTF-8 text occupies only about 5KB, whereas the corresponding Llama3-8B KV cache requires roughly 131MB. To address this issue, we further explore combining our method with KV cache compression techniques. Specifically, we consider the following choices: ❶ LLMLINGUA [12], which compresses documents by token dropping; and ❷ ANLLMS [13], which fine-tunes the LLM to compress each sentence in the document into an anchor token with special attention masks.

When using ANLLMS, we empirically observe a significant drop in LLM performance on question-answering benchmarks. To address this, we introduce the following modifications aimed at improving effectiveness. First, instead of compressing each sentence into a single anchor token and storing its KV cache, we divide each document into multiple consecutive chunks of fixed length and compress each chunk into multiple anchor tokens. Second, we modify the attention masks so that each anchor token attends only to tokens within its own chunk and to preceding anchor tokens. More specifically, let $x$ represent a training example from the pretraining dataset, consisting of a sequence of tokens. We split the first $N$ tokens into $N/s$ chunks of fixed size $s$:

$$\boldsymbol{x}^{(n)} = x_{(n-1)s+1}, \ldots, x_{ns}, \ n = 1, \ldots, N,$$

where $s$ denotes the chunk size. In our experiments, we set $s = 100$ by default. We then append several anchor tokens to each chunk. The LLM is trained to perform next-token prediction on the remaining tokens beyond position $N$, which are restricted to attend only to the anchor tokens. This setup encourages the model to compress the input into a small number of anchor tokens, allowing us to store only the KV cache of those tokens and thereby significantly reduce storage overhead. Also, by increasing the number of anchor tokens, we can better maintain the model performance given compressed KV cache. To apply this method, we first pretrain the LLM using this objective on pretraining data. We then fine-tune the model on QA datasets using KVLINK. Additional implementation details are provided in Appendix A.6.

## 3 Experiments

### 3.1 Experiment Setup

**Comparison baselines.** We evaluate our method against three existing approaches: ❶ PROMPT-CACHE [9], which directly reuses the precomputed KV cache of each retrieved document or context segment for model inference. Since the original PROMPTCACHE does not handle positional encoding mismatches, we enhance it by applying our positional re-encoding method (Section 2.2) to ensure the stored KV cache aligns correctly with the concatenated input. ❷ CACHEBLEND [6], which concatenates the precomputed KV caches of retrieved documents and then recomputes the KV cache for a small number of selected tokens within the retrieved documents. Following the original implementation, we set the recomputation ratio to 18%. ❸ BLOCKATTENTION [7], which explicitly trains the model to handle separately encoded KV caches by fine-tuning on QA tasks where retrieved documents are encoded independently. For all baselines and our method, we use Llama-3.2-1B-Instruct and Llama-3.2-3B-Instruct [15] as the backbone models. Since BlockAttention and our method require fine-tuning, we ensure fair comparison by using the same data mixture and training hyperparameters for both. Further implementation details for each baseline are provided in Appendix A.3.

**Implementation.** We adopt Llama-3.2-1B-Instruct, Llama-3.2-3B-Instruct and Llama-3.1-8B-Instruct as the backbone models, fine-tuning them for 6,000 steps using a global batch size of 64 across 8×H100 GPUs. We construct the training dataset by mixing the training sets of 2WikiMQA [16], TriviaQA [17], pretraining data from FineWeb [18], and TÜLU 3 [19]. Further details on data preprocessing, dataset mixture, and training configurations are provided in Appendix A.1 and A.2. For our method, we train three versions, each appending 0, 1, or 5 link tokens to every document or context segment, denoted as KVLINK0, KVLINK1, and KVLINK5, respectively.

**Evaluation configurations.** We evaluate the effectiveness of our method in three dimensions: ❶ performance with separately encoded KV cache, ❷ the inference efficiency, ❸ the general capability preservation (*e.g.*, math reasoning and instruction-following ability), and ❹ performance with compressed KV cache.

To evaluate the model performance when using separately encoded documents or context segments, we focus on retrieval-augmented question answering tasks including NaturalQuestions [10], 2WikiMQA [16], TriviaQA [17], HotpotQA [11], and MuSiQue [20]. For NaturalQuestions, we adopt the evaluation protocol from Liu et al. [21], where the LLM is given a question along with a set of 10 retrieved documents. The document that contains the correct answer is systematically placed at each of the 10 possible positions across separate evaluation runs. The final accuracy is reported as the average over these 10 evaluations. For 2WikiMQA, HotpotQA, and MuSiQue, we utilize the originally provided retrieved documents for evaluation. For TriviaQA we retrieve 10 documents

Table 1: Performance comparison between KVLINK and other methods. The **best** results are highlighted in **bold**. KVLINK0, KVLINK1, KVLINK5 refer to using 0, 1 and 5 link tokens respectively.

| | NQ | 2WikiMQA | TriviaQA | HotpotQA | MuSiQue | MultiNews | Samsum |
|---|---|---|---|---|---|---|---|
| *Llama-3.2-1B* | | | | | | | |
| Original Llama | 44.6% | 61.8% | 61.6% | 49.3% | 13.8% | 0.197 | 0.301 |
| Finetuned Upperbound | 46.9% | 71.9% | 68.7% | 56.5% | 19.9% | 0.193 | 0.291 |
| PROMPTCACHE | 18.6% | 19.5% | 34.9% | 20.5% | 1.4% | 0.179 | 0.199 |
| CACHEBLEND | 25.7% | 31.0% | 52.0% | 28.7% | 3.7% | 0.126 | 0.074 |
| BLOCKATTENTION | 39.0% | 64.3% | 64.6% | 48.3% | 14.3% | 0.193 | 0.247 |
| KVLINK 0 | 40.8% | 62.2% | 63.6% | 48.1% | 13.5% | 0.193 | 0.256 |
| KVLINK 1 | 43.8% | 64.9% | 65.7% | 52.9% | 16.0% | 0.194 | **0.257** |
| KVLINK 5 | **45.0%** | **66.0%** | **66.3%** | **55.6%** | **19.2%** | **0.196** | 0.256 |
| *Llama-3.2-3B* | | | | | | | |
| Original Llama | 69.4% | 60.4% | 72.6% | 69.3% | 34.8% | 0.200 | 0.328 |
| Finetuned Upperbound | 69.7% | 74.1% | 76.2% | 74.3% | 41.5% | 0.199 | 0.352 |
| PROMPTCACHE | 24.7% | 27.8% | 55.7% | 24.6% | 2.2% | 0.203 | 0.224 |
| CACHEBLEND | 42.6% | 47.6% | 64.0% | 32.7% | 5.5% | 0.191 | 0.146 |
| BLOCKATTENTION | 58.8% | 70.3% | 72.9% | 64.3% | 28.3% | **0.204** | 0.317 |
| KVLINK 0 | 62.1% | 70.0% | 73.3% | 65.9% | 28.8% | 0.203 | 0.316 |
| KVLINK 1 | 64.0% | 70.9% | 73.6% | 68.8% | 32.5% | 0.203 | 0.318 |
| KVLINK 5 | **64.4%** | **71.2%** | **73.7%** | **69.5%** | **35.8%** | **0.204** | **0.320** |
| *Llama-3.1-8B* | | | | | | | |
| Original Llama | 71.3% | 77.2% | 76.2% | 77.6% | 49.1% | 0.204 | 0.348 |
| Finetuned Upperbound | 75.1% | 78.5% | 77.9% | 78.2% | 46.5% | 0.166 | 0.353 |
| PROMPTCACHE | 28.9% | 42.2% | 62.6% | 36.5% | 6.7% | 0.203 | 0.329 |
| CACHEBLEND | 55.2% | 45.9% | 68.8% | 40.8% | 6.0% | **0.207** | 0.320 |
| BLOCKATTENTION | 70.8% | 73.6% | **76.6%** | 72.2% | 38.7% | 0.166 | 0.342 |
| KVLINK 0 | 71.0% | 73.3% | 76.4% | 72.7% | 39.9% | 0.163 | 0.342 |
| KVLINK 1 | 72.4% | **74.7%** | **76.6%** | 73.6% | 38.9% | 0.170 | 0.340 |
| KVLINK 5 | **72.5%** | 73.8% | **76.6%** | **73.8%** | **40.8%** | 0.168 | **0.345** |

using Contriever [22] following the setting in BLOCKATTENTION. In all cases, documents are encoded separately into KV cache.

Additionally, following CACHEBLEND, we evaluate on two text summarization datasets including MultiNews [23] and Samsum [24]. The LLM is given several in-context examples per instance, each separately encoded into KV cache. We report the RougeL score [25] as the metric. Notably, our evaluation datasets cover all those used by our baselines, BLOCKATTENTION and CACHEBLEND, ensuring a comprehensive and fair comparison.

To measure inference efficiency, we evaluate time-to-first-token (TTFT) under different document lengths. Specifically, we fix the number of retrieved documents at 10 and vary document lengths from 100 to 500 tokens, leading to total context lengths ranging from 1,000 to 5,000 tokens. Given pre-computed KV caches stored in CPU memory, we compare the TTFT of our method to standard LLM inference, which fully re-encodes all contexts for each query.

To ensure our method does not degrade the model's original capabilities, we evaluate it on a range of reasoning and instruction-following benchmarks, including IFEval [26], GSM8K [27], MMLU [28], ARC-Challenge [29], ARC-Easy [29], PiQA [30], SciQ [31], Winogrande [32], and HellaSwag [33]. We report the accuracy on each dataset. For IFEval, we report both strict prompt-level accuracy (IFEval-P) and instruction-level accuracy (IFEval-I). More details of the evaluation configuration are available in Appendix A.2.

Finally, to further reduce cache-storage and loading overhead, and thereby make cache reuse practical, we evaluate reuse with compressed KV caches. We report QA accuracy under various compression methods and assess the effectiveness of KVLINK when operating on the compressed caches. The only difference from the first experiment is that we use compressed, rather than original, KV cache.

## 3.2 Main Results

We first evaluate the effectiveness of our method when using separately encoded KV caches. For all QA tasks, each retrieved document is encoded into the KV cache independently. In summarization tasks, each in-context example is also encoded separately, following the same setup as our baselines. The evaluation results are presented in Table 1.

To provide a more comprehensive understanding, we also include the performance of the original Llama models and their fine-tuned versions trained on the same data mixture as other methods. Importantly, these reference models are evaluated in the standard setting, where the retrieved documents or in-context examples are concatenated and encoded as a single contiguous sequence, rather than being separately encoded into KV cache. This serves as an upper bound for performance, helping contextualize our improvements.

We highlight the key observations below. First, KVLINK consistently outperforms all baselines across all datasets, demonstrating its strong effectiveness. In most QA tasks, our approach surpasses the best baseline, BLOCKATTENTION, with up to 5% higher accuracy. The only exceptions are the 2WikiMQA and TriviaQA datasets, whose training sets are included in our training data mixture. On all other evaluation datasets, our method consistently outperforms the baselines, demonstrating superior out-of-domain generalization performance. For other baselines, the gap is more significant. The performance gap is even larger when compared to other baselines. Notably, our method not only outperforms the original Llama models without fine-tuning but also achieves accuracy close to fine-tuned Llama, which is evaluated with a fully concatenated context.

Second, the link tokens appended to each document effectively bridge separately encoded documents, restoring inter-document connections. Since one of the key differences between our method and baselines is the use of link tokens for cross-document reconnection, the observed performance improvements further validate the effectiveness of our design. With negligible computational cost, these link tokens significantly enhance performance. Compared to BLOCKATTENTION, which also fine-tunes the model on QA tasks with separately encoded documents, our method demonstrates the necessity of link tokens. Without them, BLOCKATTENTION performs worse than our approach despite using the same training data.

Finally, we observe a consistent performance gain as the number of link tokens increases, reinforcing their positive impact.

## 3.3 Inference Efficiency Evaluation

A key objective of KVLINK is to reduce the computational overhead incurred by repeatedly encoding long contexts. To demonstrate these efficiency gains, we measure the Time-to-First-Token (TTFT) when reusing the precomputed KV cache for ten documents of varying lengths. For a realistic assessment, we include the overhead of loading the precomputed KV caches onto the GPU. Each measurement is averaged over 100 runs, with an initial 10 warm-up trials to eliminate memory allocation overhead, following the setup in [34].

Figure 3 compares our methods, KVLINK1 and KVLINK5, against standard decoding, which encodes the entire context for each query. The key observations are as follows. First, reusing the precomputed KV cache significantly improves LLM inference efficiency. To leverage the precomputed KV cache, we perform three main operations: (1) loading KV caches from CPU to GPU, (2) applying new positional encoding, and (3) encoding the link tokens for each document. All of these operations introduce only negligible latency, reducing TTFT by 85%–96% compared to standard decoding.

Second, across all context sizes, both KVLINK1 and KVLINK5 consistently achieve substantially lower TTFT relative to standard decod-

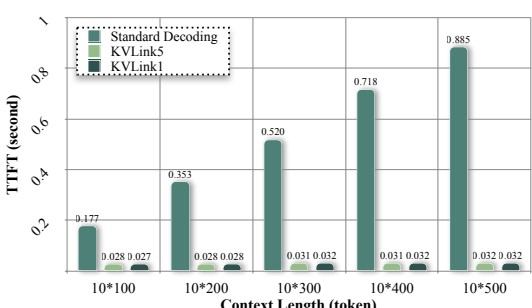

Figure 3: Inference speed comparison with ten reused contexts of varying lengths. Both KVLINK1 and KVLINK5 show considerably lower Time-to-First-Token (TTFT) than standard decoding as context size grows.

Table 2: Performance comparison between KVLINK, original Llama, and finetuned Llama under standard decoding. **WG** refers to WinoGrande, and **HS** refers to HellaSwag.

| | GSM8K | MMLU | IFEval-I | IFEval-P | ARC-E | ARC-C | PiQA | SciQ | WG | HS |
|---|---|---|---|---|---|---|---|---|---|---|
| *Llama-3.2-1B* | | | | | | | | | | |
| Original Llama | 44.9 | 46.1 | 65.9 | 54.5 | 68.5 | 35.8 | 74.3 | 93.9 | 59.5 | 45.2 |
| Finetuned Llama | 39.7 | 43.2 | 64.0 | 50.8 | 68.5 | 35.0 | 74.6 | 94.1 | 59.7 | 45.2 |
| KVLINK 0 | 41.1 | 42.0 | 63.2 | 50.1 | 67.2 | 33.5 | 74.1 | 93.5 | 57.5 | 43.7 |
| KVLINK 1 | 41.2 | 42.2 | 64.4 | 51.2 | 67.2 | 33.4 | 73.8 | 93.5 | 57.5 | 43.7 |
| KVLINK 5 | 41.0 | 42.3 | 64.3 | 51.4 | 67.0 | 33.4 | 73.8 | 93.6 | 57.6 | 43.6 |
| *Llama-3.2-3B* | | | | | | | | | | |
| Original Llama | 77.2 | 60.5 | 82.2 | 75.6 | 74.2 | 43.7 | 75.7 | 95.2 | 67.4 | 52.2 |
| Finetuned Llama | 70.0 | 60.5 | 78.4 | 69.9 | 72.3 | 40.3 | 76.2 | 96.0 | 66.8 | 51.6 |
| KVLINK 0 | 70.2 | 60.7 | 79.4 | 70.8 | 72.7 | 40.4 | 76.0 | 95.5 | 65.7 | 51.4 |
| KVLINK 1 | 70.8 | 60.9 | 79.4 | 70.4 | 72.3 | 40.0 | 76.0 | 95.6 | 65.5 | 51.4 |
| KVLINK 5 | 70.5 | 60.9 | 79.4 | 70.4 | 72.6 | 40.0 | 76.0 | 95.6 | 65.5 | 51.4 |
| *Llama-3.1-8B* | | | | | | | | | | |
| Original Llama | 85.1 | 68.0 | 85.1 | 78.7 | 81.9 | 51.7 | 79.8 | 96.7 | 73.7 | 59.1 |
| Finetuned Llama | 75.4 | 64.3 | 82.7 | 75.6 | 80.9 | 48.3 | 80.3 | 96.3 | 72.9 | 58.6 |
| KVLINK 0 | 75.2 | 64.8 | 82.8 | 75.8 | 82.1 | 50.8 | 80.0 | 96.8 | 72.5 | 58.9 |
| KVLINK 1 | 75.5 | 64.0 | 82.4 | 75.4 | 82.2 | 51.1 | 80.7 | 96.8 | 72.8 | 59.0 |
| KVLINK 5 | 75.7 | 64.0 | 83.0 | 76.2 | 81.5 | 50.4 | 80.3 | 96.6 | 72.4 | 59.1 |

ing. As the reused context length increases, the
TTFT gap further widens, highlighting the scalability of our approach. Notably, when the context length reaches 5,000 tokens, KVLINK decreases inference latency by 96%. These results confirm that KVLINK effectively mitigates the computational cost of large-scale context encoding and is well-suited for scenarios requiring fast response times.

## 3.4 General Capability Preservation

While our primary objective is to enable efficient KV cache reuse, we also verify whether these modifications maintain the model's general reasoning and instruction-following capabilities. Table 2 presents the results for three model variants: Llama-3.2-1B, Llama-3.2-3B and Llama-3.1-8B. We compare the original models, their fine-tuned counterparts, and our method with different numbers of link tokens (KVLINK0, KVLINK1, and KVLINK5).

Our key findings are as follows. First, KVLINK maintains competitive performance across all tasks, with only marginal differences from the fine-tuned models. This confirms that KVLINK successfully preserves the model's general capabilities. Across both model sizes, the results of KVLINK are highly comparable to the fine-tuned Llama, demonstrating that our method does not significantly degrade the model's general capabilities despite restructuring context encoding.

Second, while there are minor drops in certain benchmarks (*e.g.*, ARC-C and Winogrande) compared to the original models, these differences remain within a reasonable range, typically less than 3%. We expect that performance could be further improved by refining the data mixture and incorporating additional data on reasoning and instruction-following tasks, which we leave for future work.

## 3.5 KVLINK with Cache Compression

In this section, we further evaluate the effectiveness of our method when combined with KV cache compression techniques, which help reduce the overhead of storing and loading precomputed KV caches. For both compression methods described in Section 2.4, we experiment with 50% and 75% compression rates. We fine-tune the Llama-3.2-1B model using our method and the baseline models on compressed KV caches using the training sets of 2WikiMQA and TriviaQA. Additional training details are provided in Appendix A.6. We compare our method with 5 link tokens to the best-performing baseline, BLOCKATTENTION, under KV cache compression on QA tasks.

Table 3: Evaluation of KVLINK under different compression regimes. We report the QA accuracy when using compressed cache. The value to the left of the slash (/) corresponds to a 75% compression rate and the value to the right corresponds to 50% compression.

|  |  | NQ
75% / 50% | 2WikiMQA
75% / 50% | TriviaQA
75% / 50% | HotpotQA
75% / 50% | MuSiQue
75% / 50% |
|---|---|---|---|---|---|---|
| Prompt Compression | BLOCKATTENTION | 33.9 / 37.8 | 47.3 / 58.2 | 60.1 / 64.8 | 36.1 / 41.1 | 7.1 / 11.1 |
|  | KVLINK | 35.5 / 41.6 | 47.8 / 58.2 | 61.0 / 65.7 | 37.3 / 46.6 | 6.5 / 11.5 |
| Our Method | BLOCKATTENTION | 37.5 / 40.5 | 68.4 / 70.0 | 67.0 / 68.8 | 51.1 / 54.3 | 14.0 / 16.7 |
|  | KVLINK | 40.9 / 43.0 | 69.9 / 69.4 | 68.2 / 69.3 | 52.4 / 55.4 | 14.8 / 17.3 |

The evaluation results are shown in Table 3. We highlight the following findings. First, compared to LLMLINGUA, the modified ANLLMS compression retains substantially more information, resulting in consistently stronger performance across all QA benchmarks. Second, adding link tokens consistently mitigates the accuracy drop introduced by compression, restoring the model performance when using both KV cache compression techniques.

### 3.6 Ablation Study

We also perform ablation studies examining the robustness of our method against different data mixtures. Due to space constraints, the detailed results are provided in Appendix A.4. Our key observation is that our method can still achieve competitive performance given different data mixtures. Also, we find that specific combinations of tasks and domain coverage can slightly influence downstream performance, indicating the importance of balanced data selection. While our initial findings offer insight into more effective training mixtures, we make a comprehensive exploration of optimal data configurations for future work.

## 4 Related Work

Efficient inference for LLMs. As model sizes grow, serving LLMs efficiently becomes increasingly challenging. Previous work has tackled this via model pruning [35–38], quantization [39–44], and optimized decoding algorithms such as non-autoregressive decoding [45], speculative decoding [46–48], or early-exiting [49, 50]. Like these approaches, our method also aims to enhance LLM inference efficiency by reducing inference-time computation.

KV cache reuse. Recent work has explored reusing precomputed KV caches across multiple queries, often focusing on prefix-only reuse [51–53]. A few methods extend reuse to arbitrary positions, aiming for greater efficiency. PROMPTCACHE[9] allows KV cache reuse with discontinuous position IDs but ignores cross-chunk attention, causing performance drops. It also relies on Prompt Markup Language, limiting flexibility across tasks. CACHEBLEND reduces positional mismatches by selectively recomputing caches for tokens with the largest value-state variance; however, it suffers from performance degradation and the non-trivial cost of recomputing 10%–20% of tokens (plus all tokens at the first layer). BLOCKATTENTION[7] removes positional embeddings by re-encoding KV caches and directly fine-tunes on block-attention data, but lacks strategies to better connect reused contexts or refine data processing, leaving performance below ideal levels. TurboRAG [54] introduces two special tokens to mark the boundaries of reused caches. However, it still falls short in effectively reconnecting reused caches, similar to BLOCKATTENTION. A detailed discussion is provided in Appendix A.8.

KV compression. While our method focuses on accelerating inference by reusing KV caches, a complementary line of work aims to reduce memory overhead through cache eviction and quantization. For instance, [55–58] introduces an eviction policy that significantly lowers the memory footprint during generation, while [59, 42, 39, 41, 60–62] investigate quantizing KV caches to minimize storage requirements with only minor performance loss. Notably, our approach can be seamlessly integrated with these state-of-the-art eviction and quantization strategies, thereby addressing both speed and memory efficiency.

Retrieve-augmented generation. Retrieval-augmented generation (RAG) is widely used to enhance the generation capabilities of language models by retrieving supporting documents. Retrievers such

as dense retrievers or BM25 [22, 63, 64] are typically employed to find the most relevant documents for a given user query or context. In addition to simply concatenating the retrieved text with the input [65], there are also other methods for integrating the retrieved knowledge into the models. For instance, LongMem [66] encodes documents into caches and uses a trained side net to incorporate these knowledge caches into the context for long-context tasks. With the advances of LLMs across various NLP tasks, certain studies have focused on improving RAG with LLM [67, 68]. Some research also explores how to enhance the robustness of LLMs when processing retrieved knowledge. For example, RAFT [69] provides a fine-tuning strategy that strengthens LLMs against noisy contexts in domain-specific RAG.

## 5 Conclusion

In this paper, we introduced a method to improve LLM efficiency by reusing pre-computed KV caches. With precomputed KV caches for retrieved documents or context segments, our method can avoid redundant computation for overlapping contexts across different queries. In the future, we will refine our data mixture and investigate optimal fine-tuning strategies to further enhance performance. Additionally, we plan to explore real-world deployment scenarios to maximize the inference efficiency of LLMs. The limitation and societal impact sections are included in the Appendix.

## Acknowledgments

The work of Jingbo Yang, Bairu Hou and Shiyu Chang was partially supported by National Science Foundation (NSF) Grant IIS-2338252, NSF Grant IIS-2207052, and NSF Grant IIS-2302730. The computing resources used in this work were partially supported by the Accelerate Foundation Models Research program of Microsoft.

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

Table 4: Overview of tasks, datasets, and sample counts with and without context reuse.

| | Separately Encoded KV Cache | | | Standard Decoding | | |
|---|---|---|---|---|---|---|
| Task | Retrieval-aug. QA | Multi-turn conv. | Summarization | Retrieval-aug. QA | SFT | Pre-training |
| Data Source | TriviaQA, 2WikiMQA | DaringAnteater | XSum | TriviaQA, 2WikiMQA | Tulu3-sft-mixture | Fineweb |
| Percentage | 10% | 25% | 5% | 10% | 30% | 20% |
| Total # of Samples | 20,000 | 92,700 | 17,345 | 20,000 | 732,100 | 10,000,000 |

# A  Appendix

## A.1  Data Mixture

We fine-tuned our model using a mixture of different datasets.

**Retrieval-augmented QA.** Since RAG is the most directly relevant task, we first consider fine-tuning the model on QA tasks with retrieved documents that are separately encoded to ensure it can produce high-quality responses under this setting. More specifically, we sample questions from `TriviaQA` [17] and `2WikiMQA` [16]. For each question, we retrieve ten relevant Wikipedia passages using Contriever [22], each encoded separately. To obtain high-quality supervision, we prompt GPT-4 to generate reference answers, which serve as the ground truth for training.

**Multi-turn conversation.** Many existing instruction-following datasets [70] contain multi-turn conversations between the user and the assistant, which provides a natural way to train the model on disjointed contexts. We randomly convert earlier conversation turns into independently encoded KV caches, and the LLM is then trained to produce the appropriate responses in subsequent turns. This ensures the model to learn to integrate segmented contexts while maintaining its instruction-following capability.

**Summarization.** Summarization serves as another useful setting where the model must aggregate information from independently encoded document chunks to generate a coherent summary. We adopt the `XSum` [71] dataset. Each document is randomly split into multiple consecutive segments, and each segment is independently encoded into KV cache. The model is trained to generate the ground-truth summary, ensuring it learns to integrate information across separate segments.

Additionally, to better preserve the model's original capabilities, we also include a standard version of `TriviaQA` and `2WikiMQA`, where the retrieved documents are encoded as a whole rather than separately. In this setting, the model is trained to generate the ground-truth answer conditioned on the fully concatenated context. Lastly, we incorporate the T"ᴜʟᴜ 3 dataset [19] to preserve the instruction-following ability and a small portion of pre-training data from `Fineweb` [18] to preserve the language modeling capability.

Table 4 provides an overview of the data used in our training. For the multi-turn conversation data, each prior user–assistant conversation is independently encoded as a reused context, and we compute the training loss only on the assistant's final response. In the summarization task, we split the source document into 100-token segments, each serving as a reused context. All training examples are truncated to a maximum length of 4096 tokens.

## A.2  Implementation Details

**Training.** we propose fine-tuning the LLM with a mixed dataset drawn from different sources, enabling the model to integrate disjoint context segments while preserving its original capabilities.

Figure 5 illustrates the system prompts we use for each task category during training. These prompts are generated by GPT-4 [72], and are randomly picked during training. Additionally, for the QA training data, we also shuffle the retrieved reused documents in the context A.5.

Figure 4 demonstrates how each data sample is processed when the context is reused in KVLɪɴᴋ. Particularly, we also include two special tokens, `KV-START` and `KV-END` to specify the boundaries of the reused contexts. For each reused document or context segment, the link tokens are dependent on the index of the document in all the reused documents, which means, in different prompts, for

```
Multi-turn Conversation
<SYS> You are an AI assistant. </SYS> <KV_START>
<USER> Write me an email about … </USER>
<ASSITANT> Here is the email about … </ASSITANT> <link1_1> <link1_2>
<USER> The subject is not good enough … </USER>
<ASSITANT> Here is a revised version… </ASSITANT> <link2_1> <link2_2> <KV_END>
<USER> You also need to modify … </USER>
<ASSITANT> OK … </ASSITANT>
```

```
Retrieve-augmented QA
<USER> <KV_START>
Document [1] (Title: … )
<link1_1> <link1_2>
Document [2] (Title: … )
<link2_1> <link2_2> …
<KV_END>
Question: … </USER>
```

```
Summarization
<USER> <KV_START>
The Nobel Prizes are five separate
<link1_1> <link1_2>
prizes awarded to those who conferred
<link2_1> <link2_2> …
<KV_END>
Summarize the above passage. </USER>
```

Figure 4: **Data Preprocess for Context Reuse.**

**SFT**
"*You are an AI assistant. Provide helpful, accurate, and clear answers. When uncertain, explain your reasoning or request clarification.*"

"*You are an AI assistant. Focus on achieving the user's goal in each interaction. Use concise yet informative explanations.*"

"*You are an AI assistant. Speak clearly and stay consistent with prior statements. If you need more information, politely ask for it.*"

"*You are an AI assistant. Provide truthful, well-sourced information whenever possible. Acknowledge any limitations and avoid speculation if unsure.*"

**QA**
"*You are an AI assistant. Use the provided documents to answer the user's question. If the information is insufficient, acknowledge the gap or request clarification.*"

"*You are an AI assistant. Always ground your answers in the retrieved documents and do not add unsupported details. If the documents lack sufficient information, indicate that.*"

"*You are an AI assistant. Rely solely on the given documents for evidence when answering questions. When necessary, cite or paraphrase the document content accurately.*"

"*You are an AI assistant. Base your replies on the retrieved documents, ensuring completeness and correctness. Ask for more details if the documents do not cover the question fully.*"

**Summarization**
"*You are an AI assistant. Read the provided text and produce a concise summary. Capture the main points without unnecessary detail.*"

"*You are an AI assistant. Summarize the essential ideas from the given text. Avoid minor details and focus on critical insights.*"

"*You are an AI assistant. Provide a brief, high-level overview of the text. Ensure clarity and coherence, prioritizing key themes.*"

"*You are an AI assistant. Summarize the text clearly and logically. Organize the main ideas in a coherent sequence.*"

Figure 5: **System Prompts Used for Training.** We employ tailored system prompts for three primary task types—SFT, QA, and Summarization—reflecting different objectives and guiding the model's responses accordingly.

the $n$-th reused document, its link tokens are always `link`$n$`-1`, `link`$n$`-2`, ..., `link`$n$`-K`, if $K$ link tokens are used.

**Evaluation.** For QA evaluation with separately encoded KV cache, we adopt the accuracy as the evaluation metric, following [7, 21, 73, 74], which considers the prediction is correct if any sub-string of the prediction exactly matches any gold answer.

For the evaluation of general capability preservation, we use LM Evaluation Harness [75]. We use few-shot examples for `GSM8K` and `MMLU`(8-shot for `GSM8K` and 5-shot for `MMLU`).

## A.3 More Implementation Details of Baselines

For BLOCKATTENTION, we process the training data with context reuse in 4 by separately encoding each reused document or context segment and concatenating the caches directly without inserting any special tokens in between. For the data with no context reuse, the data processing is the same as KVLINK.

For PROMPTCACHE, in its original implementation, the position encoding is not adjusted when reused in the new prompt. It uses discontinuous position information between reused caches, which is not accurate. We maximize the performance of PROMPTCACHE by giving all the reused contexts with gold position information during evaluation.

## A.4 Ablation Studies on Data Mixure

Table 5: Performance on different training data mixture.

|  | NQ | 2Wiki | TriviaQA | HotpotQA | MuSiQue | Samsum | GSM8K | MMLU | IFEval-I | IFEval-P | ARC-E | ARC-C | PiQA | SciQ |
|---|---|---|---|---|---|---|---|---|---|---|---|---|---|---|
| **KVLink5** | 45.0 | 66.0 | 66.3 | 55.6 | 19.2 | 0.256 | 41.0 | 42.3 | 64.3 | 51.4 | 67.0 | 33.4 | 73.8 | 93.6 |
| **No Summarization** | 46.5 | 66.6 | 67.2 | 56.6 | 18.5 | 0.250 | 40.6 | 43.2 | 61.5 | 49.9 | 67.6 | 34.6 | 73.7 | 94.6 |
| **No Multi-turn Conv.** | 48.2 | 68.7 | 67.5 | 56.2 | 17.8 | 0.262 | 40.2 | 44.2 | 63.5 | 51.7 | 68.0 | 33.1 | 73.9 | 94.7 |
| **QA only** | 48.9 | 69.5 | 68.4 | 58.3 | 17.3 | 0.242 | 43.5 | 45.2 | 63.9 | 52.8 | 64.9 | 34.7 | 72.9 | 90.5 |

We experiment with various data mixtures under the KVLINK5 setting, each omitting a distinct subset from our original training mix (see Appendix A.1). Specifically, we explore three configurations: **No Summarization**, **No Multi-turn Conversation**, and **QA Only** (see Table 5). For the first two, we retain the same relative proportions among the remaining tasks and train for 6,000 steps, while **QA Only** is trained for two epochs to prevent overfitting. Our results show that tasks requiring cross-chunk reasoning are essential for robust cache reuse. Moreover, incorporating multiple types of cache-reuse data improves generalization. However, the optimal recipe of training data for fully equipping LLMs with general cache-reuse capabilities remains an open question, which we leave to future work.

## A.5 Impact of Answer Document Position

Empirically, we have observed that directly constructing the QA training data using Contriever [22] retrieved relevant documents is not ideal because the retriever typically places the answer-containing document near the front based on the relevance score. As a result, the fine-tuned model "learns" to find answers at the beginning of the context rather than reasoning across the entire reused context.

To validate our statement, we fine-tune the Llama-3.2-3B base model on the same training data without shuffling as BLOCKATTENTION and evaluate under the [21] setup. In [21], the test set for NaturalQuestions is also built with Contriever-retrieved contexts but places the document containing the correct answer at varying positions. As shown in Figure 6, the model's performance drops significantly whenever the ground-truth document is located further back in the context.

## A.6 Training of Modified ANLLM Compression

Our training of modified ANLLM compression contains two stages: continuous pre-training and QA tuning.

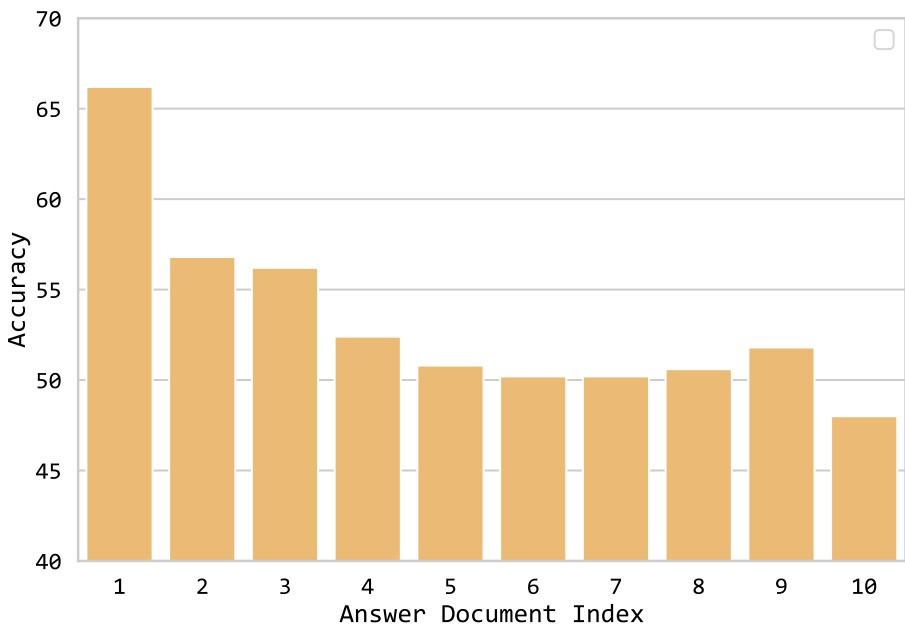

Figure 6: **Impact of Answer Document Position on Accuracy.** The accuracy substantially decreases when the correct document is located farther from the start, indicating the necessity of shuffling the retrieved documents in the QA training data.

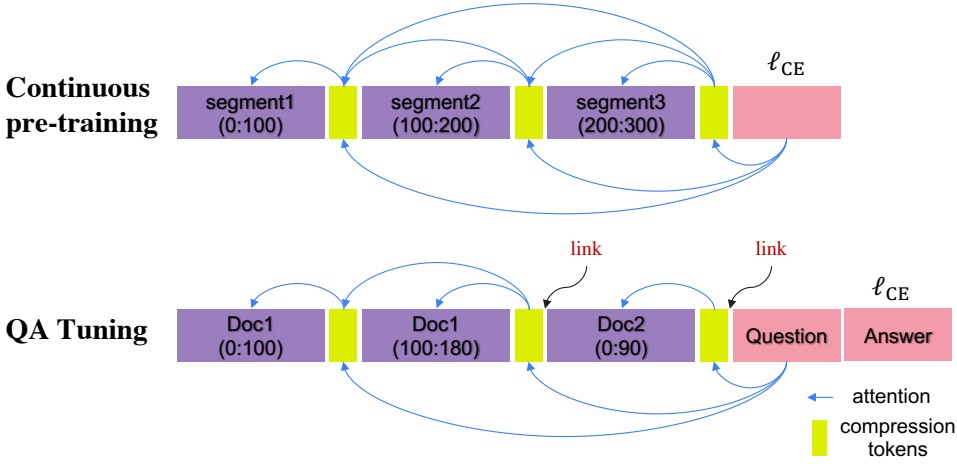

Figure 7: The attention during training compression tokens.

**Continuous pre-training.** This stage involves training the compression tokens to capture information from a longer preceding sequence. As shown in Figure 7, given a text sample, we first divide it into two halves. We compress the former half and calculate the cross-entropy loss on the latter half. During compression, we segment the former half into multiple text segments based on a fixed token length of 100 tokens. After each segment, we append several compression tokens. These compression tokens attend to all the preceding compression tokens and the original tokens in their segment, which distinguishes our approach from ANLLM. The original tokens in each segment maintain local attention, and the tokens in the latter half cannot attend to these original tokens, but only to the compression tokens.

**QA tuning.** The goal of QA tuning is to adapt the compression tokens to compress the document and further train the model to perform QA tasks based on the compressed information. The attention mechanism during QA tuning is similar to that in continuous pre-training. The only difference is that the compression tokens of each document cannot attend to one another.

## A.7 Storage Demand of KV Cache Reuse

Storing KV caches comes with a higher cost than storing the original text. Below, we discuss the trade-off between storage and efficiency and how it can be managed.

First, combining KVLink with existing KV cache compression methods helps reduce storage usage. In Section 2.4, we presents two such approaches, demonstrating that a well-designed compression strategy causes minimal performance loss. Ongoing improvements in compression can further reduce the storage demand.

Second, GPU usage cost is typically higher than storage cost. As shown in our experiments, when running a Llama3.1-8B model on an A100 80GB GPU, KVLink can cut latency by 96% for a 5000-token input. Under fixed GPU hours, this allows KVLink to handle 25 times more requests than standard decoding. For every million of these requests, KVLink takes 9 GPU hours (16 USD), while standard decoding uses 246 GPU hours (440 USD), based on the current rate (1.79 USD/hour for A100 80GB). On the other hand, storage remains inexpensive. For example, Amazon S3's standard plan charges just 0.023 USD per GB each month.

Finally, we can lower storage cost through system-level solutions. Two design strategies can be used:

❶ Cache only high-hit-rate documents. Strategies like LRU or LFU can help identify which documents to keep in cache. Others can be stored in plain text as usual.

❷ We can use tiered KV cache storage to further save storage cost. For example, although a 1,000-token document stored as UTF-8 text occupies about 5KB, whereas its Llama3-8B KV cache requires roughly 131MB. We can put this cache in cheaper storage if it is seldom used. In general, low-access caches should be saved on cheaper storage like SSDs, while important ones stay in faster memory like CPU RAM.

## A.8 Comparison with TurboRAG

Another baseline for KV cache reuse is TurboRAG [54], we include its discussion here as it is similar to another baseline in our experiments, BlockAttention [7], where the reused caches are directly concatenated. More specifically, ❶ TurboRAG introduces two extra tokens: prepending <doc_start> to the document and appending <doc_end> to the document. Similar to BlockAttention, TurboRAG computes and stores the KV cache of the document and the two tokens with local self-attention. That means the two tokens are precomputed offline as part of the document and only maintain local attention within the document to mark the document boundaries. ❷ Therefore, these two tokens are mainly used to indicate the boundaries of documents. In contrast, our method introduces link tokens and recomputes their KV cache at testing time to reconnect the separately encoded documents. ❸ During the training phase, we also introduce the link tokens and train them with the objective of connecting the separately computed contexts, which brings better performance compared to TurboRAG and BlockAttention.

We also conduct experiments to empirically compare KVLink to TurboRAG. We replicate TurboRAG and train it using the same training data and training setup as our method. Result is shown in Table 6

| | NQ | 2WikiMQA | TriviaQA | HotpotQA | Musique | avg. |
|---|---|---|---|---|---|---|
| *Llama-3.2-1B* | | | | | | |
| KVLINK5 | 45.0 | 66.0 | 66.3 | 55.6 | 19.2 | 50.4 |
| TurboRAG | 43.4 | 65.5 | 64.8 | 51.8 | 15.2 | 48.1 |
| *Llama-3.2-3B* | | | | | | |
| KVLINK5 | 64.4 | 71.2 | 73.7 | 69.5 | 35.8 | 62.9 |
| TurboRAG | 62.9 | 69.5 | 72.9 | 65.6 | 31.4 | 60.5 |

Table 6: Comparison of KVLink5 and TurboRAG across QA datasets.

### A.9 Limitations

Although our approach achieves state-of-the-art performance while improving the inference efficiency, it still has several limitations. Although KVLINK can restore performance with compressed cache, there is still some performance degradation after compression. One possible solution is to conduct larger scale training for better compression and linking. Second, the size of KV cache varies a lot, because it is partially based on the document length. Therefore, it will be hard to store these cache in the vector database. Efficiently organizing and storing the pre-computed KV cache remains a challenge for large-scale deployment.

### A.10 Societal Impact

In this paper, our primary goal is to improve LLM efficiency by reusing KV caches. Our method is designed to improve both inference efficiency and also maintain the model performance on downstream tasks. The training data used in this paper is well-known and widely used in other projects, and we verify its quality and safety to ensure that no private or sensitive information is included in the training or evaluation process. Also, The KV cache reuse approach proposed in this paper does not introduce additional risks or bias in LLMs. While we acknowledge that any LLM can have bias or potential misuse, the likelihood of risk or misuse of our proposed technique is considerably reduced.

### A.11 License

Various datasets are included for training and evaluation, and their licenses are listed below. The IFEval, NaturalQuestions, 2WikiMQA, TriviaQA, HotpotQA dataset is released under the Apache License 2.0. HumanEval, MMLU, GSM8K, HellaSwag, WinoGrande, XSum and LM-Evaluation-Harness are under the MIT License. The ARC dataset is provided under the Creative Commons Attribution Share Alike 4.0 license, and SciQ is under the Creative Commons Attribution Non-Commercial 3.0 license. The DaringAnteater dataset is under the license of Creative Commons Attribution 4.0. The MuSiQue dataset is under the Creative Commons Attribution 4.0 International license. The Samsum dataset is under the Creative Commons Attribution Non Commercial No Derivatives 4.0 license. The MultiNews dataset is under a legal agreement with LILY LAB. PiQA is licensed under the Academic Free License v. 3.0. The Tulu3-sft-mixture and fineweb datasets are under the Open Data Commons License Attribution family license. All datasets and software packages in this study are used strictly for their intended purpose—namely, training and evaluating LLMs. We confirm that no personal identifying information or offensive content appears in our materials.

### A.12 LLM Usage

We employed GPT-4 to assist with proofreading and improving clarity throughout the text. Nevertheless, the research concepts, analysis, and original writing remain fully authored by us.

