# OpenReview forum: "KVLink: Accelerating Large Language Models via Efficient KV Cache Reuse"
_NeurIPS.cc/2025/Conference — NeurIPS 2025 poster_

### Official Review · Reviewer_5Ajo · 2025-06-21

**Clarity:** 4
**Significance:** 3
**Originality:** 3
**Rating:** 4
**Confidence:** 3

**Summary:**

This paper introduces KVLink, an approach designed to reusing KV caches across docs. The problem KVLink aims to solve is due to inherent inefficiency in many LLM applications, particularly those involving RAG, where different inputs often share overlapping context. Standard LLM inference requires re-encoding the entire concatenated context for each query, leading to redundant computations.
KVLINK addresses this by precomputing the KV cache of each doc segment independently. During inference, these precomputed KV caches are concatenated, allowing the model to reuse cached representations instead of recomputing them. To counteract the performance degradation  that typically arises from independently computed KV caches. KVLink introduces positional re-encoding and trainable cross-segment link tokens.
The experiments validated KVLINK's effectiveness across seven datasets. Using 5 link tokens, it improved QA accuracy by 4%. Additionally, it significantly reduced TTFT

**Questions:**

Questions have been proposed in weakness.

**Ethical Concerns:**

["NO or VERY MINOR ethics concerns only"]

**Final Justification:**

4:  Borderline accept

Remain unresolved:  The impact of link tokens on large models (More than 7B) appears to be limited.

Solved: Clarify the applicable scenarios of KVLink, and adopt a hierarchical memory design to store the KV cache in RAG systems.

**Limitations:**

Yes

**Paper Formatting Concerns:**

Please format the citation in L197

**Quality:**

3

**Strengths And Weaknesses:**

### Strength
1. The problem of redundant computation due to repeated encoding of shared contexts in LLMs is a genuinely interesting and critical challenge.
2. This paper introduces the KVLINK approach with positional re-encoding and trainable link tokens tackle the KV cache reuse in multi docs scenarios. The trainable link tokens are an elegant solution for restoring cross-document attention.
3. The experiments demonstrate the effectiveness of KVLINK across diverse datasets and model scales (Llama-3.2-1B, Llama-3.2-3B, and Llama-3.1-8B). While link tokens provided improvement in smaller models.

### Weakness
Overall, this paper addresses the recomputation problem by introducing link tokens to capture relationships between documents to some extent. However, it still has the following weaknesses:
1. Storing documents as KV Cache remains too costly at present.
2. Based on the results shown in the paper, most of the experimental improvements are observed in 1B and 3B models, with only marginal gains for the 8B model. Can larger models (30B/70B) be evaluated to demonstrate the general applicability of link tokens?
3. In the experiments presented in the paper, many experimental designs for certain task types, such as the NQ task, do not fully demonstrate the effectiveness of link tokens.

---

> ### Author Rebuttal · Authors · 2025-07-31
>
> We sincerely thank Reviewer 5Ajo for the valuable recognition of our work and insightful questions. Your comments have been a great help in improving and improving our work. Below is a detailed response to the key points you raised.
>
> **Q1:** **Storing documents as KV Cache remains too costly at present.**
>
> **A1:** Thank you for raising this important point. We agree that storing the KV cache is more costly than storing the text. We would like to discuss this trade-off between storage overhead and efficiency, as well as the strategies to reduce the cost, as outlined below.
>
> - First, we can reduce storage overhead by combining KVLink with existing KV cache compression techniques. In our paper, we explore two different strategies and show that the performance loss is minimal with a well-designed compression scheme. We believe the compression method can be further improved to reduce the storage cost.
> - Second, GPU cost is more significant than storage cost. As shown in our experiment, serving a Llama3.1-8B model on one A100 80GB GPU, prefilling a 5000-token context with KVLink can reduce latency by about 96%. Therefore, for requests of this length, KVLink can serve 25 times more requests than standard decoding using the same amount of GPU time. For one million such requests, KVLink uses about 9 GPU hours for prefilling, which costs around 16 USD, while standard decoding takes about 246 GPU hours, costing around 440 USD (based on the current market price of ~1.79 USD/hour for an A100 80GB GPU). Meanwhile, storage is much cheaper than GPU resources. For example, on cloud storage like Amazon S3 standard tier, storing 1 GB per month costs about 0.023 USD.
>
> - Finally, we can reduce storage cost through system-level design. Specifically, two strategies can be used:
>
>   - Store the KV cache only for documents with high hit rates. Strategies like Least Recently Used (LRU) or Least Frequently Used (LFU) can help decide which documents to keep in KV cache. For less frequently used documents, we can store them as plain text.
>
>   - Use hierarchical KV cache storage. Less frequently accessed caches can be stored on cheaper storage (e.g., SSD), while more critical ones can be kept in faster memory (e.g., CPU RAM).
>
>
>
> In summary, our work focuses on speeding up LLM inference. We leave the full system design that balances efficiency and storage cost as future work.
>
> **Q2:** **Based on the results shown in the paper, most of the experimental improvements are observed in 1B and 3B models, with only marginal gains for the 8B model. Can larger models (30B/70B) be evaluated to demonstrate the general applicability of link tokens?**
>
> **A2:** Thank you for the question. Firstly, we observed that when using the larger models, it is harder to achieve further performance improvements because these models already have high absolute performance, leaving less room for gains compared to smaller models. We agree that it is important to show that the improvements hold across different model sizes. To address the concern, we have trained the KVLink5 model and our strongest baseline, BlockAttention, using Qwen-2.5-32B-instruct as the base model. The result is given below.
>
> | Task           | NQ   | 2WikiMQA | HotpotQA | TriviaQA | Musique |
> | -------------- | ---- | -------- | -------- | -------- | ------- |
> | BlockAttention | 71.7 | 94.2     | 75.3     | 84.6     | 45.3    |
> | KVLink5        | 74.4 | 95.9     | 76.9     | 84.5     | 46.6    |
>
> We highlight that our method also outperforms the best baseline, BlockAttention, on the 32B model with a 1.3%–2.7% absolute improvement. The only exception is the TriviaQA dataset. We believe the reason is that the training set of TriviaQA is used to train both methods, so their performance on in-distribution data becomes similar. These improvement numbers are consistent with the results on 1B and 3B models, showing that our method works well across different model sizes.
>
> **Q3:** **In the experiments presented in the paper, many experimental designs for certain task types, such as the NQ task, do not fully demonstrate the effectiveness of link tokens.**
>
> **A3:** The tasks that we tested in the paper for the first experiment are QA and summarization tasks. The three reasons that we are testing these tasks are:
>
> 1.  The prefill-heavy nature of these tasks. These tasks usually have much longer context and short model response, which makes them suitable to benefit from KVLink through reusing context cache.
> 2. To provide a comprehensive evaluation and comparison, these tasks are all the tasks tested by our baselines.
> 3. The multi-hop reasoning QA tasks rely more on cross attention across documents, and QA is a task that has strong demand in real-world applications.

---

> > ### Comment · Reviewer_5Ajo · 2025-08-06
> > **Follow up questions**
> >
> > I have an additional question: how does a large language model handle link tokens? Are they assigned additional token IDs?

---

> > > ### Author Response · Authors · 2025-08-06
> > >
> > > Hi, thank you for the question. Whether to use additional token IDs depends on whether the model includes reserved special token IDs for fine-tuning.
> > >
> > > - **For the LLaMA model**, we directly use the reserved special tokens provided in its original implementation. In the tokenizer config of LLaMA 3 (shown partially below), token IDs `"128002"` and `"128003"` are examples of such reserved special tokens:
> > >
> > >   ```json
> > >   "added_tokens_decoder": {
> > >       "128000": {
> > >         "content": "<|begin_of_text|>",
> > >         "lstrip": false,
> > >         "normalized": false,
> > >         "rstrip": false,
> > >         "single_word": false,
> > >         "special": true
> > >       },
> > >       "128001": {
> > >         "content": "<|end_of_text|>",
> > >         "lstrip": false,
> > >         "normalized": false,
> > >         "rstrip": false,
> > >         "single_word": false,
> > >         "special": true
> > >       },
> > >       "128002": {
> > >         "content": "<|reserved_special_token_0|>",
> > >         "lstrip": false,
> > >         "normalized": false,
> > >         "rstrip": false,
> > >         "single_word": false,
> > >         "special": true
> > >       },
> > >       "128003": {
> > >         "content": "<|reserved_special_token_1|>",
> > >         "lstrip": false,
> > >         "normalized": false,
> > >         "rstrip": false,
> > >         "single_word": false,
> > >         "special": true
> > >       }
> > >   ```
> > >
> > >   In our implementation, we use token IDs `128011` to `128211` as link tokens, corresponding to `<|reserved_special_token_3|>` through `<|reserved_special_token_203|>`.
> > >
> > > - **For models that do not have reserved special tokens**, like Qwen, we expand the vocabulary and assign new token IDs to the link tokens.

---

> ### Comment · Reviewer_5Ajo · 2025-08-01
> **Reply to rebuttal**
>
> Thank you for the response. While my concerns have been addressed, my score will remain unchanged, as the impact of link tokens on large models appears to be limited.
>
>
> Best,

---

### Official Review · Reviewer_vcib · 2025-07-02

**Clarity:** 3
**Significance:** 3
**Originality:** 2
**Rating:** 4
**Confidence:** 3

**Summary:**

The paper proposes a method called KVLink to reuse precomputed KV caches in order to alleviate the burden of recomputing the KV cache for each retrieved document. Since simple KV cache concatenation leads to significant performance degradation, the authors propose trainable “link” tokens that can aggregate information and reconstruct the interconnection between documents. These link tokens attend to both other link tokens and to their assigned documents. From an implementation perspective, pre-RoPE is employed so that the positional indices of the KV cache can be adjusted properly. Compared to existing cache-reuse methods, the proposed approach achieves superior QA performance and summarization accuracy. Furthermore, KVLink demonstrates a significant speedup in time-to-first-token (TTFT).

**Questions:**

* During training, “link” tokens take positional indices between documents; is this policy maintained at inference time, too? In other words, do link tokens possess non-consecutive positional indices in the implementation?
* In TTFT experiments (Section 3.3), on which model and GPU were the experiments conducted, and how does TTFT vary with model size?
* In Table 1, it would be helpful to indicate which methods require fine-tuning (BlockAttention, KVLink) and which do not, for clarity.

**Ethical Concerns:**

["NO or VERY MINOR ethics concerns only"]

**Final Justification:**

Most of my concerns have been addressed. Based on the discussions and other reviews, it seems that this work has clear advantages, novel points, and potential practical issues. I wish the revised version to include a discussion about pre-RoPE, practical limitations, and approaches to mitigate such challenges.

**Limitations:**

yes

**Paper Formatting Concerns:**

No issues.

**Quality:**

2

**Strengths And Weaknesses:**

**Strengths**

* The concept of using link tokens is clear. It seems that parallelism can be maintained at inference time simply by adjusting the attention map properly.
* Experiments are well-designed and detailed. The authors put an effort into fair comparisons with related work (Sections A.3, A.4). The paper evaluates KVLink across multiple tasks, including summarization. Analyses also include the GPU loading time of the KV cache in the TTFT measurements.

**Weaknesses**

* The most significant issue, which the authors also discuss, is that storing the entire KV cache demands an overwhelmingly large storage footprint. Even with speed gains, the jump from 5 KB to 131 MB (lines 157–158) is not practical. Although methods such as AnLLM have been explored, the manuscript does not specify how much space is saved versus how much performance is lost. In my opinion, even 5KB to 1MB is too much.
* While extending into the domain of RAG and KV compression is meaningful, the methodology appears similar to AnLMM. KVLink seems to expand from chunk-level to document-level granularity. The manuscript should discuss these differences in detail or provide specific improvements focused on the application (e.g., RAG).
* The on-the-fly adjustment of RoPE positions is already well known as “pre-RoPE”, in KV cache compression and quantization papers. In this perspective, the novelty of position re-embedding (rotation) is somewhat weak.

---

> ### Author Rebuttal · Authors · 2025-07-31
>
> We sincerely thank Reviewer vcib for the valuable recognition of our work and insightful questions. Your comments have been a great help in improving and improving our work. Below is a detailed response to the key points you raised.
>
> **Q1:** **Storing the entire KV cache demands an overwhelmingly large storage footprint.**
>
> **A1:** Thank you for highlighting this important concern. We acknowledge that KV cache storage is more expensive than storing plain text. Below, we address this trade-off between storage and efficiency, along with possible ways to reduce the cost.
>
> - First, storage overhead can be minimized by combining KVLink with existing KV cache compression methods. In our paper, we study two such strategies and show that with proper compression design, the performance drop is minimal. These compression methods can be further refined to lower the storage burden.
>
> - Second, the cost of GPU usage outweighs that of storage. In our experiments, serving a Llama3.1-8B model on an A100 80GB GPU, using KVLink for prefilling a 5000-token context reduces latency by around 96%. This means that for such requests, KVLink enables 25 times more requests to be served using the same GPU time compared to standard decoding. For one million such requests, KVLink requires roughly 9 GPU hours (costing 16 USD), while standard decoding needs about 246 GPU hours (440 USD), based on the current price (1.79 USD/hour for A100 80GB). Meanwhile, storage costs remain low. For instance, Amazon S3 standard tier charges about 0.023 USD per GB per month.
>
> - Lastly, storage can be further optimized at the system level. Two strategies are applicable:
>
>
>
>   - Store KV cache only for documents with high access frequency. Techniques like Least Recently Used (LRU) or Least Frequently Used (LFU) can help choose which documents to cache. Infrequently used documents can be stored as plain text.
>
>
>
>   - Use a hierarchical storage system for KV cache. Caches accessed less often can be offloaded to lower-cost storage like SSDs, while high-priority caches can be kept in fast memory such as CPU RAM.
>
>
>
> In a nutshell, this work focuses on improving LLM inference speed. Designing a complete system to optimize the efficiency–storage trade-off is an important direction for our future work.
>
> **Q2:** **While extending into the domain of RAG and KV compression is meaningful, the methodology appears similar to AnLLM.**
>
> **A2:** Thank you for the question. Here we want to highlight the differences between our compression method and the AnLLM method from three dimensions:
>
> - We include the KV compression technique to help reduce the storage overhead of pre-computed KV cache. To this extent, we explore different strategies to compress the KV cache, including prompt compression [1], our method, and AnLLM. However, we empirically found that AnLLM does not give good performance after compression. Please find the experiment results below using AnLLM for compression. The AnLLM model here is first trained with the pre-training data as described in the AnLLM paper, and then supervised fine-tuned using QA tasks for 2 epochs, the same as our method.
>
> | Task              | NQ   | 2WikiMQA | TriviaQA | HotpotQA | Musique |
> | ----------------- | ---- | -------- | -------- | -------- | ------- |
> | AnLLM (5 anchors) | 31.3 | 36.1     | 52.6     | 20.9     | 2.6     |
> | Our method (50%)  | 43.0 | 69.9     | 69.3     | 55.4     | 17.3    |
> | Our method (75%)  | 40.9 | 69.4     | 68.2     | 52.4     | 14.8    |
>
> The sub-optimal performance of AnLLM motivates us to make a few modifications to it. We acknowledge that our method is directly built upon AnLLM with slight changes to improve performance. We also mention this in the main paper (lines 160–164).
>
> - Second, the main difference between our revised compression method and the original AnLLM is the attention mechanism. In AnLLM, the compressed tokens (i.e., the anchor tokens) in the previous chunk can be attended by the following raw tokens and anchor tokens. However, in our method, to follow the cache reuse setting where each chunk is encoded separately, the compressed tokens can only be attended by the link tokens and query tokens, which follow the standard self-attention. We empirically find this attention mechanism more suitable for our application setting.
>
> - Third, as you have mentioned, the intended use cases of the two methods are also different. AnLLM focuses on dynamically dropping the generated tokens with compression during the decoding phase for long context tasks, instead of pre-computing and compressing the KV cache for reuse. In contrast, our method compresses the document in the context into a reusable KV cache for efficient prefilling.
>
> [1] Pan, Zhuoshi, et al. "LLMLingua-2: Data Distillation for Efficient and Faithful Task-Agnostic Prompt Compression." ACL (Findings). 2024.
>
> **Q3:** **The on-the-fly adjustment of RoPE positions is well known as “pre-RoPE”, in KV cache compression and quantization papers.**
>
> **A3:** Thank you for pointing this out. We acknowledge there are multiple ways of restoring the cache position information, and we want to highlight that our key contribution in this paper is proposing the link tokens for restoring the cross attention between the reused context. As discussed in the related work section, current KV reuse works also handle the position information using similar methods. For example, BlockAttention removes the position information inside the caches by rotating the key cache with a negative angle. Nevertheless, we will definitely discuss and credit the pre-RoPE techniques in our final manuscript.
>
> **Q4:** **During training, “link” tokens take positional indices between documents; is this policy maintained at inference time, too? In other words, do link tokens possess non-consecutive positional indices in the implementation?**
>
> **A4:** The use of “link” tokens is the same for both training and inference time, where the positional indices of the “link” tokens are consecutive, ranging from the position of the last token in the preceding document to the position of the first token in the next following document.
>
> **Q5:** **In TTFT experiments (Section 3.3), on which model and GPU were the experiments conducted, and how does TTFT vary with model size?**
>
> **A5:** In TTFT experiments, the Llama3.1-8B-instruct model is used with an A100 80GB GPU. As the model size increases, the reduction in TTFT latency will also increase given the same length of context and same device. This is because larger models take more time to prefill the contexts, whereas loading a larger KV to GPU memory adds only a small extra time cost. We will include this information in our final manuscript.
>
> **Q6:** **In Table 1, it would be helpful to indicate which methods require fine-tuning (BlockAttention, KVLink) and which do not, for clarity.**
>
> **A6:** Thank you for pointing this out. We will include this information in Table 1 in our final manuscript. Specifically, for the KV reuse methods, BlockAttention and KVLink require training, while PromptCache and CacheBlend are training-free methods.

---

> > ### Comment · Reviewer_vcib · 2025-08-03
> >
> > Thank you for the response. Most of my concerns have been addressed. Based on the discussions and other reviews, it seems that this work has clear advantages, novel points, and potential practical issues. I'd like to maintain my score.

---

### Official Review · Reviewer_94Ka · 2025-07-02

**Clarity:** 4
**Significance:** 4
**Originality:** 1
**Rating:** 4
**Confidence:** 4

**Summary:**

In RAG, data is retrieved then processed by the network during prefil. This work extends something like prefil cache-ing to the RAG system as a whole, where instead of retrieving text data, network KV cache is retrieved (note given retrieved docs can have different ordering, a special attention mask must be used in training).

This paper introduces a method already introduced in https://arxiv.org/abs/2410.07590. (they even intro special tokens to act as what this work calls "links").
Its a really good idea; just already exists. Not novel :(

**Questions:**

please cite: https://arxiv.org/abs/2410.07590
PLEASE rewrite your paper comparing to TurboRAG and resubmit. please please please. I would 100% accept if it was redone that way.

what is the size of your vector store vs the baseline where just text is stored (storing text much cheaper than storing kv-caches)

**Ethical Concerns:**

["NO or VERY MINOR ethics concerns only"]

**Final Justification:**

They're including discussions on how their work improves over TurboRAG.

**Limitations:**

not novel
kv-caches are larger than text. this will blow up the size of your vector store.

I'd also be good to talk about storage vs compute economics and cache coherence

**Paper Formatting Concerns:**

none I noticed

**Quality:**

4

**Strengths And Weaknesses:**

Strengths:
- really good idea
- The potential efficiency gains are f-in cracked (i'm like 80% sure the big labs do a version of this; the big labs already to disag prefil-decode; might as well stick a cache in-between the prefil and decode GPUs...)

Weaknesses:
- Not novel; missing tubrorag as related work
- Scalability and Practical Deployment Issues
  - Missing analysis of storage overhead costs vs. computational savings trade-offs
    - KV caches are massive compared to text
  - No analysis of cache management strategies for large document collections
  - No cost analysis of storage infrastructure requirements
- I'm not you test performance vs long context well (ie number of retrieved docs vs performance)

---

> ### Author Rebuttal · Authors · 2025-07-31
>
> We sincerely thank Reviewer 94Ka for the valuable recognition of our work and insightful questions. Your comments have been a great help in improving and improving our work. Below is a detailed response to the key points you raised.
>
> **Q1:** **Missing TurboRAG as related work.**
>
> **A1:** Thank you for pointing out this important work. We will include the discussion of TurboRAG in our experiments and related work section in the final manuscript. We would like to compare our method with TurboRAG as below.
>
> - First, we want to clarify that our key contribution, the link token mechanism, is different from TurboRAG. Instead, TurboRAG is very similar to our baseline, BlockAttention, where the reused caches are directly concatenated. More specifically,
>
>   - TurboRAG introduces two extra tokens: prepending doc_start to the document and appending doc_end to the document. Similar to BlockAttention, TurboRAG computes and stores the KV cache of the document and the two tokens with local self-attention. That means the two tokens are precomputed offline as part of the document and only maintain local attention within the document to mark the document boundaries.
>
>   - Therefore, these two tokens are mainly used to indicate the boundaries of documents. In contrast, our method introduces link tokens and recomputes their KV cache at testing time to reconnect the separately encoded documents.
>
>   - During the training phase, we also introduce the link tokens and train them with the objective of connecting the separately computed contexts, which brings better performance compared to TurboRAG and BlockAttention.
>
> - Second, we also conduct experiments to empirically compare our method to TurboRAG. We replicate TurboRAG and train it using the same training data and training setup as our method. Below is the result:
>
> | Task            | NQ   | 2WikiMQA | TriviaQA | HotpotQA | Musique | avg. |
> | --------------- | ---- | -------- | -------- | -------- | ------- | ---- |
> | **Llama3.2-1B** |      |          |          |          |         |      |
> | KVLink5         | 45.0 | 66.0     | 66.3     | 55.6     | 19.2    | 50.4 |
> | TurboRAG        | 43.4 | 65.5     | 64.8     | 51.8     | 15.2    | 48.1 |
> | **Llama3.2-3B** |      |          |          |          |         |      |
> | KVLink5         | 64.4 | 71.2     | 73.7     | 69.5     | 35.8    | 62.9 |
> | TurboRAG        | 62.9 | 69.5     | 72.9     | 65.6     | 31.4    | 60.5 |
>
> We observe a consistent performance improvement of KVLink over all tasks, which shows the importance of our link tokens in restoring cross attention.
>
> **Q2:** **Scalability issue of storing pre-computed KV cache.**
>
> **A2:** We appreciate your thoughtful feedback. We agree that storing KV caches comes with a higher cost than storing the original text. Below, we discuss the trade-off between storage and efficiency and how it can be managed.
>
> - First, combining KVLink with existing KV cache compression methods helps reduce storage usage. Our paper presents two such approaches, demonstrating that a well-designed compression strategy causes minimal performance loss. Ongoing improvements in compression can further reduce the storage demand.
>
> - Second, GPU usage cost is typically higher than storage cost. As shown in our experiments, when running a Llama3.1-8B model on an A100 80GB GPU, KVLink can cut latency by 96% for a 5000-token input. Under fixed GPU hours, this allows KVLink to handle 25 times more requests than standard decoding. For every million of these requests, KVLink takes 9 GPU hours (16 USD), while standard decoding uses 246 GPU hours (440 USD), based on the current rate (1.79 USD/hour for A100 80GB). On the other hand, storage remains inexpensive. For example, Amazon S3’s standard plan charges just 0.023 USD per GB each month.
>
> - Finally, we can lower storage cost through system-level solutions. Two design strategies can be used:
>
>   - Cache only high-hit-rate documents. Strategies like LRU or LFU can help identify which documents to keep in cache. Others can be stored in plain text as usual.
>
>   - We can use tiered KV cache storage to further save storage cost. For example, although a 1,000-token document stored as UTF-8 text occupies about 5KB, whereas its Llama3-8B KV cache requires roughly 131MB. We can put this cache in cheaper storage if it is seldom used. In general, low-access caches should be saved on cheaper storage like SSDs, while important ones stay in faster memory like CPU RAM.
>
> In conclusion, our paper focuses on accelerating LLM inference. We leave the challenge of designing an optimal system that balances performance and storage for future work.
>
> **Q3:** **The performance of the proposed method with different numbers of retrieved documents.**
>
> **A3:** We agree that it is crucial to test the generalizability of KVLink under different numbers of retrieved documents. To address your concern, we evaluate the performance of our method and the best baseline, BlockAttention, on the NaturalQuestions benchmark by varying the number of retrieved documents. The experiment results are as below.
>
> | Num of Docs     | 3    | 4    | 5    | 6    | 7    | 8    | 9    | avg. |
> | --------------- | ---- | ---- | ---- | ---- | ---- | ---- | ---- | ---- |
> | **Llama3.2-1B** |      |      |      |      |      |      |      |      |
> | KVLink5         | 65.6 | 62.0 | 60.8 | 54.0 | 51.4 | 49.6 | 46.4 | 55.7 |
> | BlockAttention  | 58.8 | 60.6 | 56.8 | 44.0 | 43.8 | 39.4 | 37.0 | 48.6 |
> | **Llama3.2-3B** |      |      |      |      |      |      |      |      |
> | KVLink5         | 80.0 | 74.6 | 72.6 | 72.6 | 70.2 | 68.2 | 66.8 | 72.1 |
> | BlockAttention  | 73.8 | 75.4 | 74.0 | 69.2 | 64.0 | 64.0 | 60.8 | 68.7 |
>
> Here we can observe that for almost all the numbers of retrieved documents, the KVLink model outperforms the BlockAttention model with up to 10% performance gain. This result indicates that KVLink can generalize to different numbers of retrieved documents. We will add this experiment in our final manuscript.

---

> > ### Comment · Reviewer_94Ka · 2025-08-01
> >
> > Thank you for the response
> >
> > Including TurboRAG in the related works and discussing / showing how your work improves over turboRAG is great. TY
> > I'll improve my score.

---

### Official Review · Reviewer_534y · 2025-07-03

**Clarity:** 3
**Significance:** 3
**Originality:** 3
**Rating:** 5
**Confidence:** 4

**Summary:**

The paper introduces KVLINK, a method for accelerating large language models (LLMs) by enabling efficient key-value (KV) cache reuse across queries with overlapping context segments, such as in retrieval-augmented generation (RAG) scenarios. Traditional LLMs redundantly re-encode identical context segments for each query, leading to significant computational inefficiency. KVLINK addresses this by precomputing the KV cache for each document independently and then concatenating these caches at inference time, allowing for direct reuse.

**Questions:**

1. How does KVLINK perform in scenarios where the overlap between context segments across queries is low? Is there a break-even point where the overhead of precomputing and managing caches outweighs the benefits?

2. Is there any degradation in model performance for tasks that require strong cross-document reasoning? Given, often times it's not possible to know ahead of time, what use case the query+documents are for.

**Ethical Concerns:**

["NO or VERY MINOR ethics concerns only"]

**Limitations:**

yes

**Quality:**

3

**Strengths And Weaknesses:**

[+] KV cache positional re-encoding, and trainable cross-segment special (link) tokens as techniques to overcome the performance degradation of independently encoded KV caches.

[+] Significant Efficiency Gains: KVLINK achieves up to 96% reduction in time-to-first-token latency by reusing precomputed KV caches, making it highly scalable for applications with overlapping contexts

[-] The KVLINK technique works under the assumption of the existance of the `c_{linki}` tokens. However, section 3.1 only mentions " fine-tuning them for 6,000 steps using a global batch size of 64 across 8×H100 GPUs". Does this mean that `c_{linki}` is just a special set of markers and NOT special token (OR) there is an additional mid-training phase where the tokens are baked in?

---

> ### Author Rebuttal · Authors · 2025-07-31
>
> We sincerely thank Reviewer 534y for the valuable recognition of our work and insightful questions. Your comments have been a great help in improving and improving our work. Below is a detailed response to the key points you raised.
>
> **Q1:** **Are link tokens just a special set of markers and NOT special tokens (OR) there is an additional mid-training phase where the tokens are baked in?**
>
> **A1:** Thank you for your question. These link tokens are indeed special tokens, designed to restore the cross attention between the documents at test time. Please find our explanations below.
>
> - First, we highlight that the hidden representations of these link tokens are recomputed at inference time using standard self-attention. Instead of treating them as markers to separate each document and saving their KV cache together with the documents, the KV cache of these tokens is recomputed to restore the cross attention between documents. We do not have a separate mid-training phase for link tokens. Instead, the embeddings of link tokens and all other tokens are trained together during the fine-tuning stage. Through this stage, the link tokens learn to compensate for the missing cross-attention between the separately computed KV caches.
>
> - Second, we empirically verify whether the link tokens act as special tokens and not just as markers to separate documents. We conduct another experiment where we insert two special tokens, doc_start and doc_end, as markers to indicate the boundaries of each reused document. Unlike link tokens, these two tokens are precomputed offline with the document and only maintain local attention inside the document. This method uses the same training process and computation as our method. Below is our experimental result:
>
> | Task            | NQ   | 2WikiMQA | TriviaQA | HotpotQA | Musique |
> | --------------- | ---- | -------- | -------- | -------- | ------- |
> | **Llama3.2-1B** |      |          |          |          |         |
> | KVLink5         | 45.0 | 66.0     | 66.3     | 55.6     | 19.2    |
> | Markers         | 43.4 | 65.5     | 64.8     | 51.8     | 15.2    |
> | **Llama3.2-3B** |      |          |          |          |         |
> | KVLink5         | 64.4 | 71.2     | 73.7     | 69.5     | 35.8    |
> | Markers         | 62.9 | 69.5     | 72.9     | 65.6     | 31.4    |
>
> We observe a consistent performance improvement with link tokens. This performance gap shows the necessity of using link tokens to restore cross attention.
>
> **Q2:** **How does KVLINK perform in scenarios where the overlap between context segments across queries is low? Is there a break-even point where the overhead of precomputing and managing caches outweighs the benefits?**
>
> **A2:** Thank you for this insightful question! We agree that KVLink speeds up pre-filling by reusing the cache across different queries, but when the overlap between the queries becomes low, the cost of managing the cache may outweigh the benefits it brings. One extreme example is when all queries retrieve different documents.
>
> However, we believe that the overhead in this case can be reduced in two ways:
>
> - **Tier-based caching system.** We believe the cache management cost can be further optimized through system-level design. Specifically,
>
>   - Some documents are frequently retrieved, while others are rarely reused. So it is more storage-efficient to store only the cache of documents with high hit rates. Here, strategies like Least Recently Used (LRU) or Least Frequently Used (LFU) can be applied.
>
>   - It is helpful to offload less-used caches to cheaper storage tiers and reserve fast memory for the most important ones. In practice, we can set up a hierarchical cache across different hardware: keep the most-used KV caches in GPU VRAM (for fastest reuse), move moderately-used caches to CPU RAM, and store rarely-used caches on SSD or disk.
>
> - **On-the-fly KV computation.** Instead of precomputing caches for all documents offline in advance, we can compute the cache for a document only when it is retrieved. This way, only documents that have been retrieved before are cached, which avoids overhead from managing rarely used documents. This strategy can also be combined with the tier-based caching system above as an engineering solution when deploying the RAG system.
>
> We also want to highlight that the benefit of KVLink is to save GPU cost by serving more requests with the same GPU hours, and GPU cost is usually much higher than the storage cost of the cache.
>
> **Q3:** **Is there any degradation in model performance for tasks that require strong cross-document reasoning?**
>
> **A3:** Many of our experiment tasks already require strong cross-document reasoning. In particular, 2WikiMultiHopQA, HotpotQA, MuSiQue, and TriviaQA all require the model to combine knowledge from multiple documents to produce the correct answer. Although the absolute accuracy on MuSiQue is lower because it is a harder task, we observed that for all of these tasks, our KVLink method gives a consistent performance improvement over all the baseline methods.

---

> > ### Comment · Reviewer_534y · 2025-08-07
> >
> > Thank you for the response.
> >
> > The results on the benefits with link tokens is quite interesting and worth-wile expanding on the paper.
> >
> > However, regarding your comment on "Many of our experiment tasks already require strong cross-document reasoning. In particular, 2WikiMultiHopQA, HotpotQA, MuSiQue, and TriviaQA.." I am not sure I would consider HotpotQA nor TriviaQA to require "strong corss-document reasoning". They do require multi-hop, but I wonder what feature of these 2 datasets in particular would you argue would demonstrate strong reasoning? Thank you!

---

> > > ### Author Response · Authors · 2025-08-08
> > >
> > > Thank you for your thoughtful follow-up. Please see our response below:
> > >
> > > - **HotpotQA requires a diverse set of logical reasoning processes** for LLMs to infer the correct answer from the retrieved documents. Specifically, HotpotQA classifies its questions into four main categories: **Comparing two entities**, **Inferring the bridge entity**, **Locating the answer entity by checking multiple properties**, and **Inferring the property of an entity through a bridge entity**. All four categories require the model not only to process information across multiple documents but also to perform additional reasoning to arrive at the correct answer, rather than simply summarizing information. We provide the two examples below:
> > >   - **Comparing two entities.** These questions involve comparing information from multiple sources, and often include **temporal reasoning** or **quantitative reasoning**. For example, the question “Which building is used for more different uses, MiMA or 270 Park Avenue” requires the model to find the use lists for each building and compare the number of uses for each one.
> > >   - **Inferring the bridge entity.** To solve these questions, the model must first identify the bridge entity before the next hop, enabling it to complete the reasoning chain. For instance, in the question “Lawrence Turman had produced a film, with a character named ‘Johnny 5’ that had a follow-up sequel released in what year?”, the model should identify “Short Circuit” as the bridge entity using the supporting facts: a) Turman produced Short Circuit; b) its lightning-struck robot is Johnny 5. After identifying the bridge, the model can combine the third fact, “Short Circuit 2” came out in 1988, to produce the final answer.
> > > - TriviaQA is relatively easier, and we will revise our writing to reflect this. Nevertheless, other datasets we use do require complex reasoning similar to HotpotQA. For example, **MuSiQue introduces even more challenging multi-document reasoning questions than HotpotQA** by removing reasoning shortcuts (often caused by overly specific sub-questions or insufficient distractors) and forcing models to reason through all intended hops.
> > >
> > > In our final manuscript, we will further discuss how the reasoning requirements differ among the benchmarks we adopted and explain why link tokens are particularly beneficial for enhancing cross-document reasoning. We also acknowledge that there are other benchmarks requiring reasoning capabilities in other domains, such as mathematics and coding, and we leave training and evaluation in those scenarios for future work.
> > >
> > > Thank you again for helping us improve our submission.

---

### Note · Authors · 2025-08-14

We sincerely thank all reviewers for their encouraging recognition and insightful comments. Below is a summary of our discussion and final remarks:

- **Strengths agreed by the reviewers.**

Reviewers recognized the strengths of this work, including large efficiency gains of up to 96% TTFT reduction (Reviewer 534y, 94Ka, vcib, 5Ajo), an effective and elegant solution of using trainable link tokens to restore cross-document attention (Reviewer 534y, 94Ka, vcib, 5Ajo), SOTA accuracy on QA tasks across diverse baselines (Reviewer 534y, vcib, 5Ajo), well-designed and detailed experiments (Reviewer vcib, 5Ajo), clear presentation (Reviewer 534y, 94Ka, vcib, 5Ajo).

- **Concerns raised by the reviewers & how we addressed them**

1. How to handle storage overhead. (Reviewer 534y, 94Ka, vcib, 5Ajo)

   We proposed an effective cache compression method in our original submission. In rebuttal, we added system-level cache management designs, including hierarchical KV storage and LRU/LFU strategies for frequently used document caches. We also highlighted that GPU serving costs far exceed storage costs, emphasizing the benefits of KVLink.

2. Missing comparison with related methods TurboRAG (Reviewer 94Ka), AnLLM (Reviewer vcib)

   We clarified the difference between our method with AnLLM and TurboRAG. In addition, we show that our method outperforms both baselines across all QA tasks, such as 34.5% higher accuracy than AnLLM.

3. KVLink performance v.s. Number of documents. (Reviewer 94Ka)

   We added experiments and observed KVLink introduced performance gain with all different numbers of retrieved documents on NQ benchmark.

4. Evaluation on a larger model scale. (Reviewer 5Ajo)

   We evaluated our method on a 32B model, and observed a consistent performance gain as on smaller model scales.

We truly appreciate reviewers’ and ACs’ time and effort, and will include these discussions in the final version. In this work, we introduce KVLink, a lightweight framework for efficient KV cache reuse. KVLink trains a small set of link tokens that (1) dynamically condense redundant KV entries and (2) propagate attention across cached chunks. The method preserves state-of-the-art accuracy across 7 benchmarks while enabling significant efficiency improvement and substantial GPU cost savings with up to 96% time-to-first-token reduction, establishing foundations for future research on building efficient LLM systems.

---

### Decision · Program_Chairs · 2025-09-17

**Decision:**

Accept (poster)

**Comment:**

The paper presents KVLink, a novel approach for efficient KV cache reuse in large language models, particularly benefiting applications like Retrieval-Augmented Generation. Strengths include substantial efficiency gains, with up to 96% time-to-first-token reduction, and the effective, elegant solution of trainable link tokens for cross-document attention. KVLink also achieves state-of-the-art accuracy on QA tasks and is supported by well-designed experiments across diverse model scales, including evaluation on a 32B model.

Key weaknesses included concerns about the storage overhead of KV caches, initial lack of direct comparison with related methods like TurboRAG and AnLLM, and questions regarding its general applicability to very large models.

The paper is recommended for acceptance due to its significant efficiency improvements and the novel, effective link token mechanism that addresses a critical challenge in LLM inference.

During the rebuttal period, reviewers (534y, 94Ka, vcib, 5Ajo) raised concerns about storage overhead, which authors addressed by proposing compression, hierarchical storage, and emphasizing the higher cost of GPU usage over storage. Missing comparisons with TurboRAG and AnLLM were resolved by authors providing empirical results and clarifying fundamental differences in their link token mechanism. The nature of link tokens was clarified as special tokens recomputed at inference to restore cross-attention, empirically demonstrating their necessity. Authors also provided additional experiments showing consistent performance across varying numbers of documents and on a 32B model. Reviewer 5Ajo, while acknowledging addressed concerns, noted the perceived limited impact on large models in their final comment, though authors did show consistent gains on 32B. These discussions significantly strengthened the paper's claims and justifications.